# Confidence Interval Estimation of Predictive Performance in the Context of AutoML

Konstantinos Paraschakis[1]  Andrea Castellani[4]  Giorgos Borboudakis[1]
Ioannis Tsamardinos[1,2,3]

[1]JADBio Gnosis DA S.A., 700 13 Heraklion, Crete, Greece
[2]Institute of Applied and Computational Mathematics, FORTH, 700 13 Heraklion, Crete, Greece
[3]Department of Computer Science, University of Crete, 700 13 Heraklion, Crete, Greece
[4]Honda Research Institute Europe GmbH, 63073 Offenbach/Main, Germany

**Abstract**  Any supervised machine learning analysis is required to provide an estimate of the out-of-sample predictive performance. However, it is imperative to also provide a quantification of the uncertainty of this performance in the form of a confidence or credible interval (CI) and not just a point estimate. In an AutoML setting, estimating the CI is challenging due to the "winner's curse", i.e., the bias of estimation due to cross-validating several machine learning pipelines and selecting the winning one. In this work, we perform a comparative evaluation of 9 state-of-the-art methods and variants in CI estimation in an AutoML setting on a corpus of real and simulated datasets. The methods are compared in terms of inclusion percentage (does a 95% CI include the true performance at least 95% of the time), CI tightness (tighter CIs are preferable as being more informative), and execution time. The evaluation is the first one that covers most, if not all, such methods and extends previous work to imbalanced and small-sample tasks. In addition, we present a variant, called BBC-F, of an existing method (the Bootstrap Bias Correction, or BBC) that maintains the statistical properties of the BBC but is more computationally efficient. The results support that BBC-F and BBC dominate the other methods in all metrics measured.

## 1 Introduction

In any practical application of supervised machine learning, one needs to provide an estimate of the out-of-sample (*hereafter, oos*) performance (i.e., an estimate of generalization performance) of the final model. However, it is also important to quantify the uncertainty of this performance estimate. This quantification is often presented in the form of a confidence interval, or CI hereafter for short[1]. A $a$-CI is defined as an interval that includes the true performance of the model with (at least, if being conservative) probability $a$ in identical repetitions of the analysis with new datasets from the same data distribution. In the rest of the paper, by default $a = 0.95$ and the term CI refers to a 95% CI, unless otherwise stated. Following the literature [19] of CIs of predictive performance, we focus on *one-sided CI*. In one-sided intervals the upper bound of the interval is the maximum possible performance; the lower bound of the interval is adjusted so that the the true performance falls above that level (at least) 95% of the time. In contrast, two-sided intervals "allow" some of these 5% failing cases to fall higher than the upper bound of the interval. The reason for this preference is that in practice it is important to find the tighter lower bound of performance. The importance of quantifying uncertainty is shown in this simple example: a model with 0.70 AUC could be as good as random guessing if the CI is the interval $[0.3, 1.0]$. CIs can also facilitate comparison of models: two models with respective AUC performances of 0.85 and 0.90, may actually be on par if their CIs are $[0.70, 1.0]$ and $[0.75, 1.0]$.

---

[1]In Bayesian statistics, uncertainty is quantified with credible intervals instead. For all practical purposes, both CIs and credible intervals are employed in the same way for decisions. In the rest of the paper, we will focus on CIs.

Important properties for CI estimation are the *inclusion percentage* and the interval lower bound *tightness*. For a 95% CI estimate the inclusion percentage should ideally also be 95%. If it is higher the estimate is unnecessarily wide and conservative, but still acceptable. If the inclusion percentage is lower than 95%, the estimate is optimistic and, arguably, misleading. Tightness is defined as the difference between the true performance and the lower bound of the interval. Between two CI estimates with an inclusion percentage of at least 95%, the tighter interval is the most informative (smaller positive tightness is better). Ideally, a CI estimate should cover exactly 95% of probability with the smallest tightness possible. For example, the interval [0.00, 1.00] is a 95% CI of the AUC for any binary classification task, but so conservative that is completely useless.

Accurately estimating CIs is challenging, *particularly in the context of AutoML*. In most AutoML systems, numerous machine learning pipelines (a.k.a., **configurations**) are tried and the winning one is employed to construct the final model [18]. Returning the cross-validated performance estimate and the CI of the winning configuration exhibits the "winner's curse" bias [27, 28], which can be up to 0.2 AUC [29]. Intuitively, the phenomenon of "winner's curse" can be explained as an increasing probability to overfit the test set or sets (in cross-validation) as many models are tried [17, 28].

In this paper, we examine and benchmark the state-of-the-art methods proposed for CI estimation applicable to an AutoML context. We employ JADBio [30] as our AutoML platform of choice to generate, fit, and cross-validate numerous configurations on a corpus of real binary classification datasets covering imbalanced classes and small-sample scenarios. The configurations include the application of feature selection algorithms and linear and non-linear classifiers with different values for their hyper-parameters. Experiments on real datasets are complemented with ones on synthetic datasets under controlled conditions. The oos predictions of these configurations on each sample in the cross-validation test folds are employed to select the winning configuration. The final predictive model is built using the winning configuration. The 95% CI of its performance is estimated from the matrix of oos predictions using 9 different state-of-the-art methods and variants. All algorithms employed are model agnostic; they only require the prediction matrix to provide estimates and do not depend on the inner workings of the configurations. The methods are compared and evaluated with respect to their inclusion percentage and tightness. The methods benchmarked include a new variant called BBC-F standing for Bootstrap Bias Correction on Folds. BBC-F extends the previous BBC method [27] that was shown to remove the bias due to the winner's curse and return accurate estimations of performance. However, an evaluation of BBC w.r.t. providing CI estimates was lacking.

The results demonstrate that BBC and BBC-F dominate all other methods in all metrics measured. BBC-F is on par with BBC in terms of inclusion percentage or tightness, but it is computationally more efficient. Hence, within the scope of our experiments, we would suggest BBC-F as the method of choice for CI estimation. The contributions of the paper are to extend previous evaluations to (a) all available methodologies for CI estimation, (b) extend previous evaluations to imbalanced and small-sample tasks, and (c) propose the BBC-F variant and conclude with a clear winning methodology. While informative, the evaluation still has numerous limitations. The limitations, open problems, future directions, and the impact of this work are discussed in separate sections that conclude the paper.

## 2 Bootstrap Bias Correction for Performance and CI Estimation

We now present in detail the main ideas of Bootstrap Bias Correction (BBC) and its new variant BBC Fold (BBC-F). The input to these methodologies is an oos prediction matrix produced during model training and cross-validation. Hence, we start by describing this process first. In the context of AutoML, numerous configurations, denoted by $m_i$, are trained. The configurations may include several types of algorithms (preprocessing, imputation, feature selection, and modeling) and their hyper-parameters. The choice of the algorithm for each step can also be thought of as another

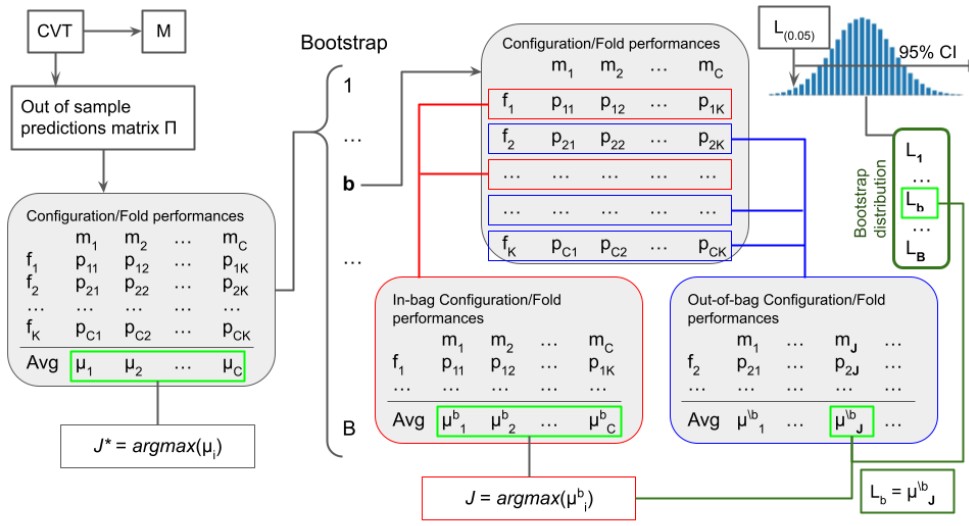

Figure 1: A schematic depiction of the BBC-F algorithm. CVT leads to an out-of-sample prediction matrix $\Pi$. $\Pi$ is used to compute the matrix $\Pi'$, containing the performance $p_{ij}$ for each configuration $m_i$ on each test fold $f_j$. The winning configuration $J^*$ is used to create the final model $M$ of CVT. Next, $\Pi'$ is bootstrapped w.r.t. to rows. In each bootstrap iteration, the winning configuration $J^b$ is selected (displayed as plain $J$ in the figure as it is itself a subscript at various places) based on the average in-bag performances. Its average performance $L_b$ on the out-of-bag performances is stored. The distribution of $\{L_1, \ldots, L_B\}$ is used to provide a point estimate (the mean of the distribution) and a CI.

hyper-parameter so that all choices of the configuration are captured in a single vector $\theta_i$. Instead of considering as having available numerous learning pipelines, we can equivalently consider that we have available only a single learning function $f$ parameterized by $\theta$ as its hyper-parameters. A static search strategy (i.e., a strategy where the set of configurations is pre-determined) in the space of all possible configurations can be encoded by a set of vectors $\theta$: $\Theta = \{\theta_i, i = 1, \ldots, n\}$. Such search strategies include grid search and random search, both of which have been employed in an AutoML context [33]. During execution, the configurations in $\Theta$ are $K$-fold cross-validated. The cross-validation procedure identifies the winning configuration according to some performance criterion (AUC, accuracy, $R^2$, c-index) which is employed to train the final model on all available data. We denote this procedure as Cross-Validation with Tuning or $CVT(f, D, \Theta)$, where $f$ is the learning method, $D$ the training data (partitioned to folds), and $\Theta$ the set of configurations to execute. The pseudo-code is in [27], omitted due to space limitations. The function call returns $\langle M, \Pi \rangle$, where $M$ is the final model, and $\Pi$ is the matrix with the predictions of each configuration on each test sample. Specifically, $\Pi_{i,j}$ contains the prediction of a model produced by configuration $m_j$ on sample with index $i$, when $i$ was in the test fold of cross-validation. Hence, $\Pi_{i,j}$ contains only oos predictions.

To produce an unbiased estimate of the performance of the final model returned by $CVT(f, D, \Theta)$ one needs to account for the fact that the winning configuration is selected among many candidates. Hence, *one needs to create an estimation procedure that produces untainted test sets that have not been used to select the winning model*. One option is to cross-validate CVT itself, i.e., cross-validate a procedure that already uses cross-validation to select the best model. This leads to the nested cross-validation protocol [21]. Each outer fold of the nested cross-validation is used only once to estimate the performance of the winning model for the corresponding training set, avoiding the winner's curse bias.

Another option is to bootstrap the CVT procedure [11]; we will call this *the direct bootstrap* approach to distinguish it from the BBC. In the direct approach, the samples of dataset $D$ are selected with replacement creating datasets $\{D^1, \ldots, D^B\}$ where $B$ is the number of bootstraps. $D^b$ are called the in-bag samples, while the samples not selected $D^{\backslash b} \equiv D \setminus D^b$ are called the out-of-bag samples. The performance $L^b$ of the winning model returned by CVT on $D^b$ is estimated on $D^{\backslash b}$. Notice that *the samples in $D^{\backslash b}$ are not used in training of the final model for dataset $D^b$, but also importantly, they have not been used to select the winning configuration either.* The average performance over all bootstraps is returned as a point estimate of performance, while the interval that covers 95% of the performance values $\{L_1, \ldots, L_B\}$ is returned as the CI. Both methodologies are computationally expensive: each of them reruns CVT and retrains $|\Theta|$ models on $K$ folds for each outer fold of the nested cross-validation or bootstrap to perform. However, *notice that neither procedure loses samples to estimation: the final model is trained on all available data by applying CVT on the original dataset* [28]. This is why nested cross-validation was employed in one of the first AutoML tools [23] that targeted low-sample, bioinformatics analyses.

The key enabling idea of BBC is to bootstrap the oos prediction matrix $\Pi$, instead of the original dataset $D$. Equivalently, instead of retraining models on each $D^b$ and selecting the winner in each bootstrap, BBC only selects the winner for each $\Pi^b$. In other words, it approximates directly bootstrapping the complete CVT procedure with bootstrapping only the step that selects the winning model, the step that creates the estimation bias. More specifically, BBC creates a bootstrap population of prediction matrices $\{\Pi^1, \ldots, \Pi^B\}$. It then identifies the winning model $M^b$ in each one of them. Assuming that the index of $M^b$ is $J^b$, an estimate of the oos performance of $M^b$ is the average performance in the out-of-bag predictions $L^b = \Pi^{\backslash b}(:, J^b)$. In the latter equation, we make use of the notation $\Pi^{\backslash b}(:, J^b)$ to denote the $J^b$ column of the matrix. From the distribution of $\{L_1, \ldots, L_B\}$ BBC can produce a point estimate of performance. The two sided $a$-CI is computed by taking the interval $[L_{(a/2)}, L_{((1-a)/2)}]$ (in the paper *higher L is considered better performance*), where $L_{(i)}$ denotes the $i$-th quantile of the distribution of $L$. For $a = 0.95$ this corresponds to $[L_{(0.025)}, L_{(0.975)}]$. The one-sided $a$-CI is computed as $[L_{(a)}, L_{max}]$.

BBC does not require training any new models, just the operations of bootstrapping a matrix and identifying the column with the maximum performance. Intuitively, we expect BBC to provide correct estimations because the winning configurations are identified from predictions on samples not used for training models in each $\Pi^b$, and the performance of the winner is estimated from samples not used to select the winner in $\Pi^{\backslash b}$. The time complexity of BBC is as follows: Given the prediction matrix $\Pi$, BBC will repeat $B$ times (for each bootstrap) the calculation of $C$ average performances, where $C$ is the number of the configurations. For simplicity, let us assume that each performance is computed in time linear to the sample size $N$ (this is true for accuracy, but AUC requires sorting the values so it has complexity $O(N \log N)$ instead). To identify the winner is a linear operation in $C$ time; checking the average performance of the winner in the out-of-bag data takes time at most $N$. The total complexity of BBC is thus $\mathcal{O}(B \cdot C \cdot N + B \cdot C)$, or simply $\mathcal{O}(B \cdot C \cdot N)$.

BBC-F is shown schematically in Figure 1; the pseudo-code is in Algorithm 1. The **BBC-F** algorithm is the same as BBC with one difference: the bootstrapping procedure of $\Pi$ does not resample over samples (rows) of the matrix but over cross-validation folds, i.e., groups of samples. To obtain BBC-F from BBC we convert the $\Pi_{N \times C}$ matrix to $\Pi'_{K \times C}$, where $K$ is the number of folds of cross-validation. Each element $\Pi'_{i,j}$ contains the average performance of a model produced by configuration $m_j$ on samples of fold $f_i$. The time complexity of the conversion of $\Pi$ to $\Pi'$ is $\mathcal{O}(C \cdot F \cdot N/F)$ (computation of $F$ average performances, each including $N/F$ samples for each configuration $C$). Hence, the total complexity of the algorithm is found by substituting $F$ for $N$ in the complexity of BBC and adding the complexity of the conversion step: $\mathcal{O}(BCF + CN)$. For a large number of bootstraps the first term dominates; the ratio of the complexity of BBC to BBC-F is in the order of $N/F$, which equals 10 for a dataset with 1000 samples being 10-fold cross-validated.

---

**Algorithm 1** BBC-F CV $(f, D = \{F_1, ..., F_K\}, \Theta)$ : Cross-Validation with Tuning, Bias removal using the BBC-F method

---

**Input**: Learning method $f$, Data matrix $D = \{\langle xj, yj \rangle\}_{j=1}^N$ partitioned into approximately equally-sized folds $F_i$ , set of configurations $\Theta$

**Output**: : Model $M$, Performance point estimation $L_{BBC-F}$, $(1 - \alpha) \cdot 100\%$ one-sided confidence interval $[b_l, b_u]$

1: $\langle M, \Pi \rangle \leftarrow \boldsymbol{CVT}(f, D, \Theta)$          ▷ Notice: the final Model is the one generated by CVT
2: Convert $\Pi_{N \times C}$ matrix to $\Pi'_{K \times C}$ by computing the performance of a configuration on a given fold.
3: **for** $b = 1$ to $B$ **do**
4:      $\Pi^b \leftarrow$ sample with replacement $K$ rows of $\Pi'_{K \times C}$
5:      $\Pi^{\backslash b} \leftarrow \Pi' \setminus \Pi^b$          ▷ Obtain the out-of-bag samples in $\Pi$ and not in $\Pi^b$
6:      $J^b \leftarrow \arg\max_j(avg(\Pi^b(:, j)))$          ▷ Select the best performing configuration on average
7:      ▷ Use min instead if lower values imply better performance.
8:      $L_b \leftarrow avg(\Pi^{\backslash b}(:, J^b))$      ▷ Estimate performance of the selected configuration from its out-of-bag fold performances
9: **end for**
10: $L_{BBC-F} = \frac{1}{B} \sum_{b=1}^B L_b$          ▷ Point estimate of average performance of $M$
11: $[b_l, b_u] = [L_{(\alpha)}, L_{\max}]$          ▷ Use $[b_l, b_u] = [L_{(\alpha/2)}, L_{(1-\alpha/2)}]$ for a two sided interval
12: **return** $\langle M, L_{BBC-F}, [b_l, b_u] \rangle$

---

BBC-F follows the same principles as BBC to ensure unbiased estimations while being more computationally efficient. On the other hand, a bootstrapping of a matrix with 1000 rows, one for each sample, will turn into bootstrapping a matrix with 10 rows, one for each fold, potentially losing information. The comparative evaluation examines the trade-off between the quality of estimation and computational complexity.

## 3 Related work

The topic of providing a point estimate of the predictive performance of machine learning models has been studied in the context of supervised machine learning. Typical methodologies include the hold-out, repeated hold-out, and $k$-Fold Cross Validation protocols [28]. Some recent works include [14, 32] and we do not attempt a full review of the literature.

Arguably, the first method for providing point estimates of predictive performance in the context of model selection or AutoML where numerous configurations are tried is the nested-cross validation protocol [21, 1, 22]. The first method to try to remove the bias of estimation due to the winner's curse is the Tibshirani and Tibshirani method (TT) [26]. The TT method did not require nesting the cross-validation step, but it did not provide accurate estimations either [29]. Nevertheless, it did inspire future work in bias removal of the winner's curse.

One of the first works for providing confidence intervals, instead of just point estimations, of performance in the context of AutoML was the Bootstrap Bias Correction Cross-validation (BBC) method [27]. The MABT [19] was recently proposed for the same problem. In addition, several other methodologies that implicitly deal with CI estimation in the presence of multiple models have been proposed or are easily adapted. We include these works as they have also been compared against MABT: these are the Tilted Bootstrapping (BT) [7], Hanley-McNeil (HM) [9], and DeLong (DL) [24], in both the standard and *10p* variation, where an auxiliary two-step selection process is employed [19]. While all methods take the same input (the oos prediction matrix, labels, and fold indices), the MABT, DL, HM and BT methods follow a different model selection process than BBC and BBC-F. We note that they have several drawbacks compared to BBC: (a) they don't return performance point estimates, (b) apply only to binary classification tasks, while BBC is metric agnostic and can be computed for other tasks, such as multi-class classification, regression or time-to-event analysis, and (c) the *10p* variation doesn't perform well on problems with low sample size or imbalanced classes, due to the two-step configuration selection process. They also require the user to set the

| | Name | Instances | Training Samples | Features | Classes | Balance ratio | Reference |
|---|---|---|---|---|---|---|---|
| *Small Sample Size* | credit-g | 1000 | 50 | 21 | 2 | 0.300 | [5] |
| | spambase | 4601 | 50 | 58 | 2 | 0.394 | [10] |
| | musk | 6598 | 50 | 168 | 2 | 0.154 | [4] |
| | phoneme | 5404 | 50 | 6 | 2 | 0.293 | [6] |
| | phishing | 11055 | 50 | 31 | 2 | 0.443 | [15] |
| *Large Sample Size* | electricity | 45312 | 500 | 9 | 2 | 0.425 | [8] |
| | mozilla4 | 15545 | 500 | 6 | 2 | 0.329 | [12] |
| | nomao | 34465 | 500 | 119 | 2 | 0.286 | [3] |
| | adult | 48842 | 500 | 15 | 2 | 0.239 | [2] |
| | bank-marketing | 45211 | 500 | 17 | 2 | 0.117 | [16] |
| | eeg-eye-state | 14980 | 500 | 15 | 2 | 0.449 | [20] |

Table 1: Benchmark Datasets used in the experiments.

proportions to split the dataset to validation/evaluation sets. The only hyper-parameter of BBC and BBC-F is the number of bootstraps to run which is intuitively easy to decide.

## 4 Experimental Set up

In the experiments, we compare BBC-F to several algorithms for CI estimation, in terms of the quality of the estimated 95% confidence intervals of the area under the ROC curve (AUC). Furthermore, we also compare the running times and scaling behavior of BBC-F and BBC. The algorithms were compared on simulated and real-world data. We only focus on low-sample settings, as (a) it has been shown that the bias of the uncorrected estimate by CVT approaches zero regardless of the number of configurations [27] (i.e., it's less than 1% for 1000 samples), and (b) CI estimates are mainly useful in low-sample settings where performance estimates have high variance (i.e., in the limit one would expect the CI range to converge to zero). We use two metrics to compare the algorithms: *inclusion percentage* and *tightness*. For real data, we use the performance on the holdout set as an estimate of true performance, which is large enough to ensure an accurate estimate. Next, we provide additional details about the algorithms, data and protocols used in the experiments.

**Algorithms and Implementations**. We compare BBC-F against two state-of-the-art algorithms for confidence interval estimation, the original BBC algorithm [27], and MABT [19]. We also compare them to standard approaches: Tilted Bootstrapping (BT) [7], Hanley-McNeil (HM) [9] and DeLong (DL) [24], both with the standard and the *10p* variation [19]. For MABT, BT, HM and DL we used the implementation of [19] available at `https://github.com/pascalrink/mabt-experiments`. As a baseline, we also directly bootstrapped the performance estimates of the selected model only, ignoring the "winner's curse" problem, referred to as Naive Bootstrapping (NB). For the comparisons on the real-world data we used JADBio [30] as the CVT method to generate and execute several configurations to run and compute the prediction matrices $\Pi$. JADBio is a commercial software-as-a-service for AutoML; it uses various feature selection methods, such as SES [13] and LASSO [25], and modeling algorithms, such as logistic regression, support vector machines, decision trees and random forests. Implementations of BBC-F and BBC, as well as all JADBio results (fold indices, configuration predictions, and outcome labels) and code for reproducing the results are available at `https://github.com/kparaschakis/BBC_algorithm`.

**Real Data**. We use datasets from OpenML [31], shown in Table 1. The datasets were selected to contain at least 1000 samples, to have enough samples for accurate performance estimation.

**Generation of Simulated Data**. We consider the following settings: sample size $N \in \{50, 500\}$, number of candidate configurations $C \in \{100, 500\}$, majority class balance $b \in \{0.1, 0.5\}$, and number of folds set to $F = \min\{10, N_{minority}\}$. First, for given values of $N$ and $b$, an outcome $Y$ is sampled from a Bernoulli distribution. Then, AUC performances are sampled for all configurations

| Config. parameters | | | Methods | | | | | | | | | |
|---|---|---|---|---|---|---|---|---|---|---|---|---|
| $(\alpha, \beta)$ N M | | b | BBC | BBC-F | DL | HM | BT | DL10p | HM10p | BT10p | MABT | NB |
| (24,6) 500 100 | | 0.1 | **0.99** (0.07) | **0.98** (0.07) | 0.90 (0.06) | 0.86 (0.04) | **0.93** (0.08) | 0.54 (0.00) | 0.26 (-0.02) | 0.75 (0.03) | 0.81 (0.04) | 0.55 (0.00) |
| | | 0.5 | **1.00** (0.04) | **0.98** (0.04) | 0.91 (0.04) | 0.90 (0.03) | **0.95** (0.05) | 0.79 (0.02) | 0.49 (0.00) | **0.93** (0.04) | **0.93** (0.04) | 0.75 (0.01) |
| 500 | | 0.1 | **1.00** (0.06) | **0.98** (0.07) | 0.86 (0.05) | 0.83 (0.03) | 0.9 (0.06) | 0.31 (-0.01) | 0.12 (-0.03) | 0.68 (0.01) | 0.82 (0.03) | 0.42 (0.00) |
| | | 0.5 | 0.98 (0.03) | **0.98** (0.03) | 0.88 (0.03) | 0.88 (0.03) | **0.94** (0.04) | 0.67 (0.01) | 0.32 (-0.01) | **0.97** (0.04) | **0.98** (0.04) | 0.69 (0.00) |
| 50 100 | | 0.1 | **0.99** (0.31) | 0.92 (0.32) | - | - | - | - | - | - | - | - |
| | | 0.5 | **1.00** (0.16) | **1.00** (0.20) | 0.73 (0.14) | 0.72 (0.10) | - | 0.10 (-0.09) | 0.06 (-0.10) | - | - | - |
| 500 | | 0.1 | **0.97** (0.32) | **0.93** (0.35) | - | - | - | - | - | - | - | - |
| | | 0.5 | **1.00** (0.17) | **0.97** (0.21) | 0.79 (0.17) | 0.77 (0.13) | - | 0 (-0.11) | 0 (-0.11) | - | - | - |
| (9,6) 500 100 | | 0.1 | **0.97** (0.09) | **0.98** (0.09) | 0.83 (0.06) | 0.70 (0.03) | 0.86 (0.07) | 0.71 (0.03) | 0.31 (-0.03) | 0.82 (0.06) | 0.83 (0.06) | 0.67 (0.01) |
| | | 0.5 | **0.98** (0.05) | **0.96** (0.05) | 0.89 (0.05) | 0.84 (0.04) | **0.93** (0.06) | 0.91 (0.04) | 0.57 (0.00) | **0.95** (0.06) | **0.95** (0.06) | 0.79 (0.01) |
| 500 | | 0.1 | **0.97** (0.09) | **0.97** (0.09) | 0.89 (0.08) | 0.84 (0.05) | 0.91 (0.10) | 0.61 (0.02) | 0.24 (-0.03) | 0.82 (0.06) | 0.85 (0.06) | 0.62 (0.01) |
| | | 0.5 | **0.99** (0.04) | **0.99** (0.05) | 0.92 (0.04) | 0.87 (0.03) | **0.96** (0.05) | 0.88 (0.03) | 0.39 (-0.01) | **0.97** (0.06) | **0.96** (0.06) | 0.74 (0.01) |
| 50 100 | | 0.1 | **1.00** (0.43) | **0.98** (0.46) | - | - | - | - | - | - | - | - |
| | | 0.5 | **0.99** (0.22) | **0.98** (0.25) | 0.82 (0.19) | 0.75 (0.12) | - | 0.43 (-0.04) | 0.2 (-0.11) | - | - | - |
| 500 | | 0.1 | **0.99** (0.42) | **0.95** (0.44) | - | - | - | - | - | - | - | - |
| | | 0.5 | **1.00** (0.22) | **0.99** (0.25) | 0.74 (0.16) | 0.70 (0.10) | - | 0.04 (-0.18) | 0.01 (-0.19) | - | - | - |
| Avg Rnk | | | 1.00 | 2.13 | 5.13 | 6.63 | 3.38 | 8.00 | 10.00 | 5.00 | 5.00 | 8.50 |

Table 2: Inclusion percentages (closer to 95% is better) and tightness (lower is better) on the simulated data. Bold numbers indicate that we don't reject the hypothesis that the value is at least 0.95.

from a $Beta(\alpha, \beta)$. In our experiments, we used $(\alpha, \beta) \in \{(9, 6), (24, 6)\}$, which correspond to mean performances of 0.6 and 0.8 with variances of 0.015 and 0.0052 respectively. Given an outcome $Y$ and the AUC of a configuration, a prediction $X_0$ is drawn from $\mathcal{N}(0, 1)$ when $Y = 0$, and a prediction $X_1$ from $\mathcal{N}(\mu, 1)$ when $Y = 1$. Then:

$$X_0 \sim N(0, 1), \quad X_1 \sim N(\mu, 1) \quad \Rightarrow \quad z := \frac{X_1 - X_0 - \mu}{\sqrt{2}} \sim N(0, 1).$$

$$AUC = P(X_1 > X_0) = P\left(\frac{X_1 - X_0 - \mu}{\sqrt{2}} > -\frac{\mu}{\sqrt{2}}\right) = P\left(z < \frac{\mu}{\sqrt{2}}\right) = \Phi\left(\frac{\mu}{\sqrt{2}}\right) \Rightarrow \mu = \sqrt{2}\Phi^{-1}(AUC),$$

where $\Phi(\cdot)$ is the CDF of the standard normal distribution, which determines $\mu$ unequivocally.

**Evaluation Protocol**. For the simulated data, each combination of settings was repeated 200 times to generate prediction matrices and compute CIs using all methods. For the real data, we performed 100 repetitions of a train/hold-out split. For small datasets (see Table 1), 50 samples were randomly sampled as the training set, while 500 were used for the larger datasets. For each split, JADBio was used (with 10-fold CV and a grid search of 766 configurations) to get the oos predictions and their fold index. Finally, each method is applied on these results, and the winning configuration is then applied on the hold-out set to get an estimate of its theoretical performance. Note, that a potentially different "winner" configuration can be selected by each of the three groups of methods: {BBC, BBC-F}, {DL, MH, BT}, and {DL10p, MH10p, BT10p, MABT}.

## 5 Experimental Results

In this section, we present the results of the experiments. We compare BBC-F to all other algorithms on simulated and real data in terms of inclusion percentage and tightness, as well as the running times of BBC-F and BBC. We also include results showing the hold-out performance of models selected by each method, as they use different model selection methods. Exact binomial tests are performed to test whether the inclusion percentage matches the theoretical CI range. Furthermore, we note that some methods failed to execute on some datasets, particularly in cases with a low-frequency minority class or extra validation/evaluation partitions.

| Dataset | BBC | BBC-F | DL | HM | BT | DL10p | HM10p | BT10p | MABT | NB |
|---|---|---|---|---|---|---|---|---|---|---|
| credit-g | **0.95** (0.24) | **0.95** (0.22) | 0.86 (0.23) | 0.69 (0.07) | 0.11 (-0.23) | 0.55 (0.02) | 0.09 (-0.19) | 0.21 (-0.10) | 0.19 (-0.12) | 0.37 (-0.05) |
| spam | **0.97** (0.19) | **0.93** (0.17) | 0.80 (0.20) | 0.80 (0.19) | 0.84 (0.15) | 0.37 (0.02) | 0.33 (-0.03) | **0.95** (0.19) | **0.93** (0.17) | 0.38 (-0.02) |
| musk | **0.97** (0.01) | **0.93** (0.00) | - | - | - | - | - | - | - | - |
| phoneme | **0.93** (0.22) | **0.91** (0.20) | 0.87 (0.27) | 0.87 (0.27) | 0.37 (-0.08) | 0.59 (0.09) | 0.41 (-0.06) | 0.54 (0.00) | 0.52 (-0.02) | 0.28 (-0.04) |
| phishing | **0.96** (0.13) | 0.90 (0.11) | 0.64 (0.15) | 0.64 (0.12) | 0.89 (0.20) | 0.24 (0.00) | 0.22 (-0.03) | **1.00** (0.28) | **0.94** (0.26) | 0.44 (-0.01) |
| elec | **0.99** (0.05) | **0.97** (0.05) | **0.97** (0.07) | **0.97** (0.06) | **0.99** (0.07) | **0.99** (0.10) | **0.94** (0.05) | **1.00** (0.13) | **0.99** (0.08) | 0.90 (0.02) |
| mozilla4 | **0.97** (0.03) | **0.94** (0.03) | 0.85 (0.04) | 0.82 (0.03) | **0.91** (0.06) | 0.84 (0.06) | 0.71 (0.02) | **0.98** (0.10) | **0.91** (0.06) | 0.76 (0.00) |
| nomao | **0.97** (0.02) | **0.97** (0.02) | 0.85 (0.02) | 0.89 (0.02) | **0.95** (0.03) | 0.88 (0.02) | 0.78 (0.01) | **1.00** (0.05) | **1.00** (0.04) | 0.70 (0.00) |
| adult | **1.00** (0.04) | **0.99** (0.04) | 0.91 (0.05) | 0.94 (0.06) | 0.91 (0.06) | **0.95** (0.07) | **0.92** (0.04) | **0.98** (0.10) | **0.95** (0.06) | 0.66 (0.01) |
| bank | **0.93** (0.06) | **0.93** (0.06) | 0.82 (0.07) | **0.92** (0.09) | 0.87 (0.08) | 0.89 (0.08) | 0.83 (0.06) | **0.94** (0.11) | **0.91** (0.11) | 0.59 (0.00) |
| eeg-eye | **0.92** (0.05) | **0.93** (0.06) | 0.87 (0.07) | 0.88 (0.06) | **0.91** (0.08) | **0.99** (0.11) | 0.79 (0.04) | **0.98** (0.13) | **0.97** (0.10) | 0.83 (0.03) |
| Avg Rnk | 1.6 | 1.8 | 5.9 | 5.3 | 5.6 | 6.9 | 8.0 | 5.8 | 4.9 | 9.0 |

Table 3: Inclusion percentages (closer to 95% is better) and tightness (lower is better) on the real data. Bold numbers indicate that we don't reject the hypothesis that the value is at least 0.95.

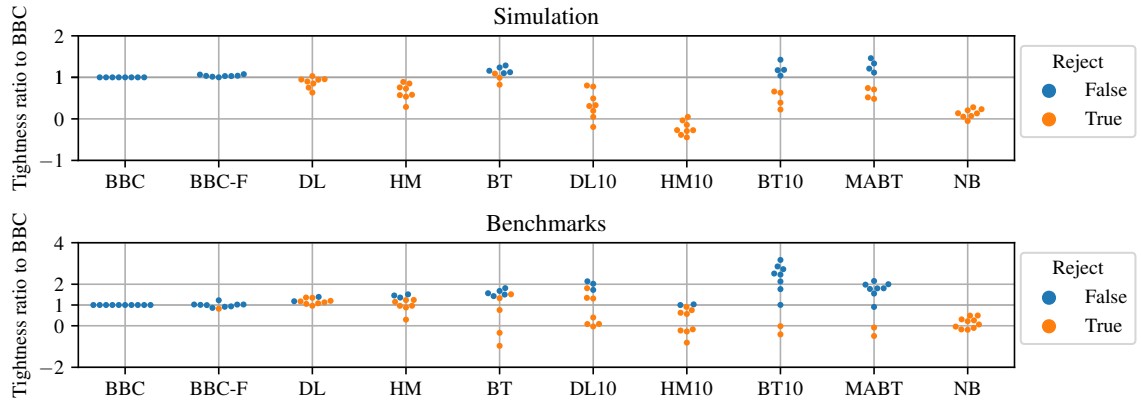

Figure 2: Relative tightness ratio to BBC per dataset (closer to 1 is more similar to BBC). Blue/orange dots correspond to non-rejected/rejected inclusion percentages.

**Comparison of CI estimation**. Tables 2 and 3 show the results for the simulated data and real data respectively. The results on the real data are split into two groups, low sample size datasets ($N = 50$) and high sample size datasets ($N = 500$). The tables show the inclusion percentages and the tightness (in parentheses) for each method and settings. Bold numbers indicate values that were not statistically significantly lower than 95% at a 5% statistical level according to an exact binomial test. The average rank of each method is shown at the bottom of the tables. For a fair comparison, only rows where all methods have returned results were used. To rank methods, the following rules are applied: (a) non-rejected (bold) methods get ranked higher than rejected ones, (b) tightness is used as a tie breaker for non-rejected methods, with lower values being ranked higher, (c) rejected methods are ranked according to their distance of their inclusion percentage to 95%, and (d) all tied methods get assigned the highest rank.

In addition to the tables, we also visually summarize the results on the swarm plots in Figure 2. Blue dots correspond to cases where the hypothesis test is not rejected, while orange dots correspond to the rejected ones. Again, only rows where all methods returned results were considered. The y-axis shows the ratio of the average tightness of each method relative to BBC, that is, a value > 1 means that the method is worse than BBC and vice versa. We chose BBC as the baseline as it was the best performing method, which also happens to always return non-rejected estimates.

The results show that BBC and BBC-F clearly dominate all competing methods, producing accurate and tight CIs. As expected, NB fails to produce accurate CIs, as it ignores the bias introduced

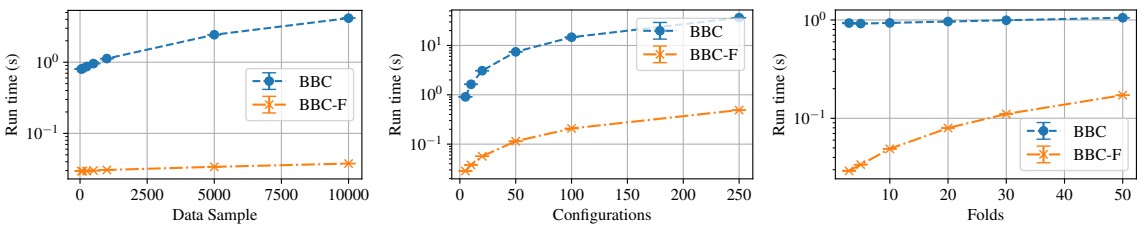

Figure 3: Time-complexity analysis for BBC-P and BBC-F wrt. (left) data samples, (mid) number of model configurations, (right) number of folds.

due to the "winner's curse". The other methods often produce statistically significantly different CIs or even fail to run, especially on smaller and unbalanced datasets and, when they don't, their average tightness is worse than that of BBC and BBC-F.

**Comparison of selected model performances**. Next, we computed the average true and hold-out performance of the selected models on the simulated and real data respectively. On the simulated data, {BBC, BBC-F} had a performance of 0.9081, followed by {DL, HM, BT} with 0.9022, and {DL10p, HM10p, BT10p, MABT} with an AUC of 0.8948. Similarly, on the real data {BBC, BBC-F} had a performance of 0.8570, followed by {DL10p, HM10p, BT10p, MABT} with 0.8499, and {DL, HM, BT} with an AUC of 0.8496. While the differences are small, BBC and BBC-F again consistently outperform all competitors.

**Comparison of BBC-F and BBC running times**. Finally, we performed an empirical evaluation to compare the running time of BBC-F and BBC. By default, sample size was set to 500, the number of configurations to 5 and the number of folds to 3. We performed 3 experiments, one for each of the above parameters, varying one of them and keeping the rest fixed, to investigate how they scale with sample size, number of configurations and number of folds.

The results are summarized in Figure 3. The y-axis shows the median running time on logarithmic scale based on 100 repetitions of the experiment. We observe that BBC-F consistently outperforms BBC by 1-2 orders of magnitude. Additionally, both algorithms exhibit similar scaling behavior w.r.t. the number of configurations, while the running time of BBC-F and BBC is not affected by sample size and number of folds respectively, as expected (see discussion in Section 2). It is important to note that, in practical applications, the number of folds is typically no higher than 10. Overall, BBC-F performs almost identically to BBC, while being computationally faster.

## 6 Impact, Limitations, and Conclusions

After careful reflection, the authors have determined that this work presents no notable negative impacts on society or the environment. We hope the work to have a positive scientific impact raising awareness regarding the information to provide users of machine learning and facilitating decision-making based on ML results. There are numerous limitations in the study. First, the approximation of BBC and BBC-F to direct bootstrapping is only valid for static HPO strategies. While they can be applied to dynamic strategies in principle (as they only require the prediction matrices as input), it is unclear if and how the dynamic hyper-parameter search introduces any bias, as the input dataset is used to guide its search. Other metrics of classification performance (accuracy, F1, balanced accuracy) need to be considered. BBC-F runs only when a sufficient number of folds have been cross-validated. Other supervised tasks, such as multi-class classification, regression and censored time-to-event analyses need to be considered.

The paper presents the first extensive evaluation of CI estimation methods on binary classification tasks in the context of AutoML. It introduces a new variant, BBC-F that achieves a speedup of $N/F$ vs. BBC, where $N$ is the number of samples, and $F$ is the number of folds, with a minimal drop in estimation quality. BBC and BBC-F dominate all other methods w.r.t. probability coverage and

interval tightness. The problem of computing CIs when providing performance estimates in ML has not been sufficiently studied, in our opinion. The results and limitations point to new research directions and future work in the field.

**Acknowledgements**. The research project was co-funded by the Stavros Niarchos Foundation (SNF) and the Hellenic Foundation for Research and Innovation (H.F.R.I.) under the 5th Call of "Science and Society" Action – "Always Strive for Excellence – Theodore Papazoglou" (Project Number: 9578.). Furthermore, Ioannis Tsamardinos gratefully acknowledges the financial support from Honda Research Institute Europe GmbH (HRI-EU).

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

## A  Appendix

### A.1  ROC-AUC performance

In Table 4 are reported the ROC-AUC performance on the simulation setting. In Table 5 are reported the AUC-ROC lower bound on the real-world benchmarks datasets.

### A.2  JADBio search space

- **Preprocessing**: Mean Imputation, Mode Imputation, Constant Removal, Standardization.

- **Feature Selection**: SES(maxK = {2, 3}, alpha = {0.01, 0.05}), Univariate feature selection with Benjamini–Hochberg correction(alpha = {0.001, 0.01}, maxVars = {100}), LASSO(penalty = {0.5, 1, 1.5}).

- **ML algorithms**: Decision Tree(MinLeafSize = {2, 3, 4}, pruning alpha = {0.01, 0.05}), Random Forest(n_trees = {100, 500}, MinLeafSize = {2, 3, 4}, vars_to_split = {0.816, 1, 1.154, 1.291} * sqrt(n_variables), n_splits = {1}, alpha = {1}), Logistic Regression(lambda = {0.1, 1, 10}), SVM with polynomial kernel(cost = {0.01, 0.1, 1, 10}, gamma = {0.01, 0.1, 1, 10}, degree = {2, 3}), SVM with Gaussian kernel(cost = {0.01, 0.1, 1, 10}, gamma = {0.01, 0.1, 1, 10}), SVM with linear kernel(cost = {0.01, 0.1, 1, 10}).

Table 4:

| | Config. parameters | | | | True AUC | Models lower | | | | | | | | |
|---|---|---|---|---|---|---|---|---|---|---|---|---|---|---|
| $B(\alpha,\beta)$ | N | M | b | C | | BBC | BBC-F | DL | HM | BT | DL10p | HM10p | BT10p | MABT |
| (24,6) | 500 | 100 | 0.1 | 2 | 0.929 | 0.859 | 0.856 | 0.865 | 0.885 | 0.845 | 0.915 | 0.937 | 0.891 | 0.882 |
| | | | | 5 | 0.806 | 0.728 | 0.728 | - | - | - | - | - | - | - |
| | | | 0.5 | 2 | 0.936 | 0.900 | 0.898 | 0.898 | 0.903 | 0.889 | 0.909 | 0.928 | 0.890 | 0.887 |
| | | | | 5 | 0.806 | 0.754 | 0.753 | - | - | - | - | - | - | - |
| | | 500 | 0.1 | 2 | 0.945 | 0.881 | 0.878 | 0.892 | 0.906 | 0.877 | 0.946 | 0.962 | 0.920 | 0.903 |
| | | | | 5 | 0.808 | 0.754 | 0.753 | - | - | - | - | - | - | - |
| | | | 0.5 | 2 | 0.954 | 0.924 | 0.922 | 0.925 | 0.927 | 0.916 | 0.934 | 0.952 | 0.908 | 0.903 |
| | | | | 5 | 0.809 | 0.779 | 0.779 | - | - | - | - | - | - | - |
| | 50 | 100 | 0.1 | 2 | 0.873 | 0.562 | 0.555 | - | - | - | - | - | - | - |
| | | | | 5 | 0.804 | 0.501 | 0.478 | - | - | - | - | - | - | - |
| | | | 0.5 | 2 | 0.902 | 0.737 | 0.700 | 0.729 | 0.772 | - | 0.975 | 0.983 | - | - |
| | | | | 5 | 0.805 | 0.576 | 0.564 | - | - | - | - | - | - | - |
| | | 500 | 0.1 | 2 | 0.877 | 0.560 | 0.529 | - | - | - | - | - | - | - |
| | | | | 5 | 0.804 | 0.507 | 0.495 | - | - | - | - | - | - | - |
| | | | 0.5 | 2 | 0.913 | 0.747 | 0.707 | 0.706 | 0.752 | - | 1 | 1 | - | - |
| | | | | 5 | 0.806 | 0.588 | 0.574 | - | - | - | - | - | - | - |
| (9,6) | 500 | 100 | 0.1 | 2 | 0.854 | 0.764 | 0.762 | 0.777 | 0.808 | 0.760 | 0.809 | 0.863 | 0.778 | 0.773 |
| | | | | 5 | 0.617 | 0.639 | 0.639 | - | - | - | - | - | - | - |
| | | | 0.5 | 2 | 0.860 | 0.810 | 0.810 | 0.808 | 0.819 | 0.799 | 0.806 | 0.843 | 0.787 | 0.786 |
| | | | | 5 | 0.616 | 0.667 | 0.666 | - | - | - | - | - | - | - |
| | | 500 | 0.1 | 2 | 0.887 | 0.799 | 0.798 | 0.801 | 0.829 | 0.784 | 0.850 | 0.901 | 0.813 | 0.802 |
| | | | | 5 | 0.620 | 0.657 | 0.657 | - | - | - | - | - | - | - |
| | | | 0.5 | 2 | 0.900 | 0.857 | 0.854 | 0.855 | 0.863 | 0.846 | 0.852 | 0.892 | 0.824 | 0.823 |
| | | | | 5 | 0.618 | 0.690 | 0.692 | - | - | - | - | - | - | - |
| | 50 | 100 | 0.1 | 2 | 0.789 | 0.360 | 0.324 | - | - | - | - | - | - | - |
| | | | | 5 | 0.605 | 0.377 | 0.370 | - | - | - | - | - | - | - |
| | | | 0.5 | 2 | 0.820 | 0.601 | 0.576 | 0.547 | 0.614 | - | 0.805 | 0.873 | - | - |
| | | | | 5 | 0.608 | 0.458 | 0.4450 | - | - | - | - | - | - | - |
| | | 500 | 0.1 | 2 | 0.809 | 0.386 | 0.367 | - | - | - | - | - | - | - |
| | | | | 5 | 0.613 | 0.394 | 0.390 | - | - | - | - | - | - | - |
| | | | 0.5 | 2 | 0.851 | 0.635 | 0.600 | 0.586 | 0.647 | - | 0.972 | 0.982 | - | - |
| | | | | 5 | 0.613 | 0.471 | 0.467 | - | - | - | - | - | - | - |

Table 4: Simulation results: Average lower ROC-AUC performance bound of selected models.

| Dataset | Hold Out | BBC | BBC-F | Val. | Eval. | DL | HM | BT | DL10p | HM10p | BT10p | MABT |
|---|---|---|---|---|---|---|---|---|---|---|---|---|
| credit-g | 0.643 | 0.404 | 0.420 | 0.620 | 0.627 | 0.391 | 0.55 | 0.850 | 0.607 | 0.820 | 0.726 | 0.744 |
| spambase | 0.896 | 0.703 | 0.730 | 0.889 | 0.874 | 0.687 | 0.703 | 0.744 | 0.857 | 0.907 | 0.680 | 0.700 |
| musk | 1 | 0.986 | 1 | - | - | - | - | - | - | - | - | - |
| phoneme | 0.755 | 0.533 | 0.552 | 0.730 | 0.730 | 0.464 | 0.455 | 0.805 | 0.642 | 0.791 | 0.735 | 0.750 |
| phishing | 0.939 | 0.807 | 0.831 | 0.926 | 0.934 | 0.777 | 0.810 | 0.726 | 0.939 | 0.964 | 0.653 | 0.670 |
| electricity | 0.859 | 0.811 | 0.811 | 0.857 | 0.856 | 0.791 | 0.793 | 0.783 | 0.754 | 0.807 | 0.726 | 0.772 |
| mozilla4 | 0.950 | 0.920 | 0.919 | 0.950 | 0.950 | 0.908 | 0.920 | 0.894 | 0.895 | 0.933 | 0.853 | 0.895 |
| nomao | 0.979 | 0.962 | 0.962 | 0.979 | 0.979 | 0.959 | 0.959 | 0.950 | 0.957 | 0.957 | 0.930 | 0.942 |
| adult | 0.890 | 0.851 | 0.851 | 0.887 | 0.890 | 0.840 | 0.827 | 0.830 | 0.822 | 0.850 | 0.790 | 0.829 |
| bank | 0.866 | 0.805 | 0.804 | 0.867 | 0.870 | 0.801 | 0.778 | 0.786 | 0.789 | 0.816 | 0.764 | 0.761 |
| eeg-eye | 0.791 | 0.739 | 0.727 | 0.789 | 0.787 | 0.719 | 0.725 | 0.711 | 0.682 | 0.748 | 0.659 | 0.684 |
| har | 0.663 | 0.536 | 0.544 | - | - | - | - | - | - | - | - | - |
| optdigits | 0.929 | 0.747 | 0.753 | - | - | - | - | - | - | - | - | - |
| pendigits | 0.857 | 0.686 | 0.700 | - | - | - | - | - | - | - | - | - |
| CIFAR10_s | 0.605 | 0.617 | 0.612 | - | - | - | - | - | - | - | - | - |

Table 5: Average AUC-ROC lower bound results on real-world benchmarks datasets.

