# OpenReview forum: "Confidence Interval Estimation of Predictive Performance in the Context of AutoML"
_automl.cc/AutoML/2024/Conference — AutoML 2024_

### Official Review · Reviewer_mtuV · 2024-03-24

**Potential Impact On The Field Of Automl Rating:** 2
**Technical Quality And Correctness Rating:** 1
**Clarity Rating:** 3

**Summary Of Contributions:**

The main contributions of the paper are:
(1) introducing a variation of BBC, called BBC Fold, which works like BBC but on fold scores (averages) instead of per sample/instance;
(2) a comparison of nine methods to estimate the one-sided 95% confidence intervals for ROC AUC for 15 real-world datasets and 16 synthetic datasets;
(3) evidence that BBC outperforms all other compared methods w.r.t. inclusion percentages.

**Actions Required To Increase Overall Recommendation:**

While I am aware that it is likely impossible to accommodate most of my requests within the rebuttal period, I will mention all points here nevertheless.

* The comparison must include relevant baselines that are applicable in AutoML.
* Missing related work must be discussed, and the submission should be put into the context of such related work.
* The experimental setup should be overhauled, or the author should clearly reason why the chosen setup (e.g., training sample size) is relevant to the AutoML field.
* Confounding factors such as overfitting and the search spaces require ablation studies.
* The code should be overhauled to enable the reproducibility of work.

**Clarity:**

* Line 23: To improve clarity, it might be advisable to also state here why one needs an estimate of the oos performance. We need an estimate of the generalization performance, which we obtain by oos performance.
* Line 29: "%95" is a typo, I think.
* Line 60f.: It might be helpful to clarify what the test fold, which is mentioned here, refers to: inner cross-validation inside the AutoML system or outer cross-validation from an evaluation setup.
* Even at line 102, it is unclear to me what predictions the authors are referring to. Does the oos matrix consist of labels or prediction probabilities?
* Line 205ff.: Which search space was exactly used for this study? This needs to be part of the appendix.
* Line 262f.: Here the authors state that "high sample size cause issues". I cannot follow this as for the given dataset, this is not a high sample size. Moreover, BBC-F should be invariante to higher sample sizes as it only works on fold scores.
* Line 275 f.: "the approximation of BBC and BBC-F to direct bootstrapping is only valid for static HPO strategies. Adaption to Bayesian Optimization or other dynamic strategies is not trivial." - Is this also the case for other methods? And why would this break down if the bias is higher as a result of BO?

**Overall Review:**

## Positive
* The research topic is important as the current ecosystem of the AutoML field lacks support for conformal predictions.
* The proposed method BBC-F is valuable because it is a straightforward and reasonable extension of BBC.
* The paper is well-written and clear in most parts.

## Negative
* The work is missing very important related work and baselines, as highlighted above.
* The evaluation might be unfair for the compared methods, and it's unclear if confounding factors such as overfitting or search space are the main reasons for the final results.
* The evaluation might not hold up for the broader application in AutoML because it focuses on a too narrow set of experiments.
* The reproducibility of the work based on the provided code is questionable.

**Potential Impact On The Field Of Automl:**

AutoML systems regularly do not employ confidence interval estimations so far. Thus, work like this might have a high impact if they would start to employ it.

However, I am unsure how far this particular work would be built upon and not other existing related work (more on this later).
Moreover, the evaluation setting in the paper might also make its adaption less likely.

**Reproducibility:**

From going through the code (mostly to determine the search space), I seriously doubt that the work with the code as it was at submission time is reproducible. Note that the text links to https://anon-github.automl.cc/ without a specific repository (Line 209), so I am basing my comments on the uploaded zip on OpenReview.

The code is missing install instructions and documentation and presumably contains admin credentials for an API. The submission checklist mentions such flaws, and if this work is to be accepted, it definitely needs to be completely overhauled during the rebuttal.

Lastly, I am a bit confused as to why JADBio was picked to perform model selection for this work and not an open-source AutoML system or simple search space using Hyperopt or Optuna. I found no reasons for this choice in the text. This dependency seems to unnecessarily make the code much harder to use and reproduce.

**Review Confidence:**

4

**Review Rating:**

4

**Review Summary:**

I recommend the rejection of the submission at this time.

While I find the work important and would like to see more like this, this submission's approach to studying confidence interval estimation for AutoML has several flaws.

With the current state of the submission, I am not convinced that the proposed approach and the baseline methods are compared fairly or even more useful in the context of AutoML than existing conformal prediction frameworks.

**Technical Quality And Correctness:**

I have several doubts regarding the decisions made by the authors.
The following covers the doubts step-by-step.

## Missing Related Work & Missing Baselines!
While this work focuses on CI estimations in the context of AutoML, intending to remove the "winner's curse", it seems to completely ignore the large body of literature and methods that focus on general-purpose CI estimations. Specifically, the body of conformal predictions is ignored.

The submission does not mention existing tools like [MAPIE](https://mapie.readthedocs.io/en/latest/index.html) or the collection of previous work like https://github.com/aangelopoulos/conformal-prediction.

While these types of previous work may not provide CI estimations with the goal of removing the winner's curse, they may still be entirely suitable for use within AutoML and obtain the same effect as trying to remove the winner's curse.

Especially for the evaluation setup used within this work, they are applicable and might outperform the compared methods. To explain, one could also use CVT to select the final model (like in BBC and BBC-F but unlike in all other compared methods) and then only estimate the CI for the final model.
This approach would be applicable since the final evaluation only cares about how conformal the estimates are for the selected model (and not the whole HPO process).

## Goal of Application Domain of the Experimental Setup?

It is unclear what the authors are trying to simulate with their experimental setup.
The authors selected a set of real-world datasets and created another set of synthetic datasets.
Then, a tiny subset of the samples is used for training (always $\leq 5$%) of the available real-world data was used for training.
This experimental setup seems unique compared to the traditional evaluation scenarios in AutoML or conformal predictions.

Why were such small training sample sizes used, and how do they confound the results?
The authors mention only that the holdout set is "large enough to ensure an accurate estimate."
But this could also have been achieved using larger datasets and more than $5$% of the samples for training.

Moreover, because so little data was used, it is unclear whether there are significant confounding effects due to overfitting. I expect a high confounding effect as the number of configurations is higher than or equal to the number of training data points.


 ## Unfair Comparison?

The best-performing methods, BBC and BBC-F, of which the authors proposed BBC-F, seem to have an entirely different concept of bias removal than the baseline methods.

BBC and BBC-F do not adjust the model selection but return the best model found by cross-validation (see Algorithm 1). Both only adjust the returned scores/CIs.
Yet, from my understanding and as mentioned in the submission, all compared methods adjust the model selection to remove the winner's curse from the estimated scores/CIs and the selected model.

As a result, we are not comparing the approaches apples-to-apples, especially depending on the search space. Assuming there are models with higher or lower variance in predictions/generalization (let's say an SVM vs. a Random Forest as in the search space), then for BBC and BBC-F, we might estimate how conformal RFs are, while for all other methods, we estimate how conformal SVMs are.
That such a difference exists is even mentioned in the results in the paragraph about hold-out performance (Line 265 ff.).

This makes it even more important to compare to conformal prediction baselines. At the same time, this seems to indicate that we can ignore the winner's curse as we get better performance if we do so (i.e., only select based on CV score).

## Generalizability of the Comparison for AutoML!
The experimental setup provides the first evidence of the difference in performance of the compared methods.
However, there seem to be several confounding factors ignored in this study (or at most pointed out as limitations in the conclusion).
This is a significant flaw for me as it heavily biases the results.

AutoML aims to solve diverse tasks with diverse approaches, and thus, we need to know how methods react across tasks and approaches.

 To name a few factors that need to be analyzed when comparing CI methods for AutoML:
* (relative) dataset size (or dataset properties in general),
* the impact of the search space,
* the impact of the robustness/variance of the algorithms in the search space,
* the metrics (Line 277 f.),
* the HPO method (Line 276 f.).

## Other Aspects

* On the Bias Removal of BBC-F: what is the benefit of BBC-F (Algorithm 1) for bootstrapping w.r.t. the scores of all other non-selected models? How would it perform if we just bootstrap the fold scores of M? I can imagine a scenario where you have some very strong models for a subset of folds to throw off the estimation procedure for M entirely.
* Several of the references (e.g., Line 93, [34]; Line 172, [1,23]) seem somewhat arbitrary and incomplete to me. The related work is missing some prior work on the concepts of improving estimates of predictive performance by improving cross-validation. To name a few:
	* S. Dudoit, M.J. van der Laan, Asymptotics of cross-validated risk estimation in model selection and performance assessment, U.C. Berkeley Division of Biostatistics Working Paper Series, Working Paper 126, February 5, 2003.
	* T. Scheffer, Error estimation and model selection, Ph.D. thesis, Technischen Universität Berlin, School of Computer Science, 1999.
	* Ng, Andrew Y. "Preventing" overfitting" of cross-validation data." ICML. Vol. 97. 1997.
	* (also mentions nested cross-validation) Browne, Michael W. "Cross-validation methods." Journal of mathematical psychology 44.1 (2000): 108-132.
* What method did you use to obtain prediction probabilities for SVMs to compute ROC AUC, or did you use the decision function? In either case, the assumption of order import for ROC AUC in the decision function or prediction probabilities is not necessarily representative of such small training data. Does this not negatively affect BBC?
* The rankings' definition seems arbitrarily constructed (Line 239 ff.). Please provide at least a raw ranking per tightness and inclusion scores in addition to the constructed ranking method. On that note, why is inclusion more important than tightness for the experimental setup?
* For the dataset size compared in this paper, all compared methods should be extremely quick. Hence, how do the results reflect the argument that BBC-F might be quicker than BBC, and what is the impact of this in practice? In other words, how fast are the compared methods?

---

### Official Review · Reviewer_o5jP · 2024-03-25

**Potential Impact On The Field Of Automl Rating:** 3
**Technical Quality And Correctness Rating:** 3
**Clarity Rating:** 3

**Summary Of Contributions:**

The paper evaluates the estimation of confidence intervals for predictive performance in AutoML, focusing on the challenge posed by the "winner's curse" - a bias introduced by selecting the best-performing model among several candidates based on cross-validation. It comprehensively assesses nine state-of-the-art CI estimation methods, including a novel variant called Bootstrap Bias Correction-Folds (BBC-F), across various datasets. BBC-F enhances the computational efficiency of the Bootstrap Bias Correction method while maintaining its statistical properties.

**Actions Required To Increase Overall Recommendation:**

While the authors thoroughly listed the limitations and future challenges of the work, it might be beneficial to discuss how to tackle these exactly. This might include giving ideas on extending BBC-F to non-static HPO strategies and other ML tasks.

**Clarity:**

The paper is well-structured and written. The methodology is detailed, allowing for reproducibility.

**Overall Review:**

Pros:
- The approach is comprehensively evaluated across various CI-computation methods and datasets with varying degrees of imbalance.
- The BBC-F variant aims to improve computational efficiency without compromising accuracy.
- A thorough discussion of the implications for AutoML, highlighting the study's practical significance, is given.
- The approach's limitations are discussed thoroughly, including further steps and challenges.

Cons:
- Some multi-class challenges could be further elaborated, especially regarding the methods' limitations and the observed performance discrepancies.
- While the authors address this already, the static HPO focus of the paper is quite limited.

**Potential Impact On The Field Of Automl:**

The author's insights are relevant to the AutoML community, emphasizing the importance of accurately estimating the uncertainty of model performance. By comparing CI estimation methods and introducing BBC-F, the paper aims to guide users in selecting robust techniques for performance evaluation. It has the potential to contribute to more reliable and interpretable AutoML solutions.

**Review Confidence:**

4

**Review Rating:**

8

**Review Summary:**

The paper introduces BBC-F, a variant to enhance BBC’s computational efficiency and work around the “winner’s curse.” The methods are applied across a spectrum of datasets to encompass multi-class, imbalanced, and small-sample scenarios, thus evaluating the robustness of each technique. The proposed approach benefits energy-efficient evaluation for uncertainty quantification. However, it is (currently) limited to static HPO scenarios and thus not applicable to Bayesian Optimization, commonly used in AutoML. In summary, the authors provide a beneficial and interesting AutoML approach. I therefore recommend accepting the paper.

**Technical Quality And Correctness:**

The paper extends the exploration of CI estimation methods to challenging scenarios like multi-class, imbalanced, and small-sample tasks. The experimental design covers a broad spectrum of datasets and simulation settings, ensuring the findings' generalizability, limited by their scope. Overall, the paper is of sound technical quality.

---

### Official Review · Reviewer_d7J2 · 2024-03-29

**Potential Impact On The Field Of Automl Rating:** 3
**Technical Quality And Correctness Rating:** 4
**Clarity Rating:** 3

**Summary Of Contributions:**

The contributions of this paper are:
1. Comparison of all state-of-the-art CI estimation methods.
2. Extending CI benchmarking to multi-class, imbalanced and small-sample datasets.
3. Proposing a computationally efficient extension to a previous method for CI estimation.

**Actions Required To Increase Overall Recommendation:**

I would like to see empirical numbers on the time complexity of BBC vs. BBC-F. The authors discuss how their variant has a lower time complexity but do not show how much faster it is in practice.

Without this, it is harder to see why we should use BBC-F in practical AutoML pipelines.

**Clarity:**

The paper has an overall good quality of writing, and manages to clearly describe a rather compicated algorithm with several nested components.

Some minor comments on the text:
1. In the abstract and introduction, there is no need to write "CI interval". Replace with "confidence interval" or just "CI".
2. Fix the title of Fig. 2-right to spell Benchmarks correctly.
3. The caption of Fig. 1 could be made more clear, and minor grammatical errors could be fixed.
4. I believe that the authors sometimes use "a=0.95" and "a=0.05" to mean the same thing.

**Overall Review:**

This paper tackles the "winner's curse" problem, which is especially prominent in AutoML due to the large quantity of model comparisons. They perform a large scale evaluation of 9 different CI estimation methods on multiple real data scenarios, and they introduce a new variant of a CI estimation method.

It is not clear if the computational cost of BBC really is a problem for real-world use. It is discussed how its time complexity is higher for larger datasets but it is not shown empirically.

The results from the evaluation shows that the BBC-F performs almost as well as BBC, indicating that the loss of information in BBC-F is often not too damaging. This means it can be a useful method if the practical benefits of its computational speed-up are shown.

**Potential Impact On The Field Of Automl:**

The results, the new method and the recommendations in this paper have the potential to improve future AutoML work, by improving CI estimation and continuing to highlight the "winner's curse" problem.

It can serve as a robust way of quantifying the expected performance of a selection model across AutoML pipelines, such as HPO, NAS etc.

**Review Confidence:**

4

**Review Rating:**

7

**Review Summary:**

This paper tackles the "winner's curse" problem, which is especially prominent in AutoML due to the large quantity of model comparisons. They perform a large scale evaluation of 9 different CI estimation methods on multiple real data scenarios, and they introduce a new variant of a CI estimation method. I think this is a good paper that could be improved further by showiing the empirical benefits of their faster method.

**Technical Quality And Correctness:**

I believe that the experiments in this paper are of high quality due to the breadth of datasets considered, the high number of repetitions, and statistical testing. The use of a commercial system for running cross-validation gives further faith in the correctness of the overall experimental pipeline.

---

### Meta-Review · Area_Chair_ikSC · 2024-04-23

**Paper Recommendation:** Accept
**Confidence:** 3

**Metareview:**

This paper investigates the estimation of confidence intervals for predictive performance in AutoML. Throughout the rebuttal period, the authors diligently addressed reviewers' concerns. After thoroughly considering the feedback from all reviewers and the authors' responses, I recommend acceptance and strongly advise the authors to integrate the reviewers' feedback into the final version of the paper.

---

### Decision · Program_Chairs · 2024-04-29

**Decision:**

Accept

**Comment:**

Thank you for submitting your paper. We are happy to tell you that we accept your paper to the main track. See you in Paris.